# Onvansertib-Based Second-Line Therapies in Combination with Gemcitabine and Carboplatin in Patient-Derived Platinum-Resistant Ovarian Carcinomas

**DOI:** 10.3390/ijms26125708

**Published:** 2025-06-14

**Authors:** Federica Guffanti, Ilaria Mengoli, Francesca Ricci, Ludovica Perotti, Elena Capellini, Laura Sala, Simone Canesi, Chu-Chiao Wu, Robert Fruscio, Maya Ridinger, Giovanna Damia, Michela Chiappa

**Affiliations:** 1Laboratory of Gynecological Preclinical Oncology, Experimental Oncology Department, Istituto di Ricerche Farmacologiche Mario Negri IRCCS, 20156 Milan, Italy; federica.guffanti@marionegri.it (F.G.); ilaria.mengoli@marionegri.it (I.M.); francesca.ricci@marionegri.it (F.R.); ludovica.perotti@marionegri.it (L.P.); elena.capellini@marionegri.it (E.C.); michela.chiappa@marionegri.it (M.C.); 2Mouse & Animal Pathology Laboratory (MAPLab), Fondazione UniMi, 20139 Milano, Italy; laura.sala2@unimi.it (L.S.); simone.canesi@unimi.it (S.C.); 3Department of Veterinary Medicine and Animal Sciences, University of Milano, 20122 Lodi, Italy; 4R&D Department, Cardiff Oncology, San Diego, CA 92121, USA; cwu@cardiffoncology.com (C.-C.W.); mridinger@cardiffoncology.com (M.R.); 5UO Gynecology, Fondazione IRCCS San Gerardo dei Tintori, 20900 Monza, Italy; robert.fruscio@unimib.it; 6Department of Medicine and Surgery, University of Milan-Bicocca, 20126 Milan, Italy

**Keywords:** platinum resistance, ovarian carcinoma, onvansertib, gemcitabine, carboplatin, plk1

## Abstract

Platinum resistance represents an urgent medical need in the management of ovarian cancer. The activity of the combinations of onvansertib, an inhibitor of polo-like kinase 1, with gemcitabine or carboplatin was tested using patient-derived xenografts of high-grade serous ovarian carcinoma resistant to cisplatin (DDP). Two PDX models were selected from our xenobank: one with acquired resistance to DDP (#266R) and the other (#315) with intrinsic DDP resistance. Tumor-bearing mice were randomized to receive vehicle, single onvansertib, gemcitabine and carboplatin, and their combinations. Onvansertib/gemcitabine and onvansertib/carboplatin combinations were well tolerated. In the #266R model, single drug treatments were completely inactive, while the combinations of onvansertib/gemcitabine and onvansertib/carboplatin resulted in a significant increase in survival compared to controls and single drugs (*p* < 0.001 versus control, onvansertib, gemcitabine and carboplatin). Similar efficacy was observed in the s.c. #315 PDX model; indeed, onvansertib and carboplatin monotherapies were inactive, gemcitabine monotherapy was marginally active, while both combinations were highly active. The molecular mechanism underlying the efficacy of the combinations suggests a higher induction of DNA damage which seems plausible considering that, in both cases, gemcitabine and carboplatin, respectively, interfere with DNA metabolism and induce alkylation damage. The results suggest that the combinations of onvansertib/gemcitabine and onvansertib/carboplatin are safe and were shown to be of therapeutic value in the platinum-resistant setting of ovarian carcinoma, strongly supporting their clinical translatability.

## 1. Introduction

High-grade serous ovarian carcinoma (HGSOC) is the most common and lethal gynecologic malignancy. Standard treatment includes cytoreductive surgery, followed by platinum-based chemotherapy. Most patients respond to first-line chemotherapy, but almost invariably, 75% of women will relapse with a much less platinum-sensitive tumor [1]. In recent decades, anti-angiogenic drugs and poly-ADP-ribose polymerase inhibitors (PARPis) have been approved as maintenance therapy in HGSOC and have been shown to improve progression-free survival and overall survival [2]. Despite the approval of these new drugs, resistance to platinum-based therapy, both intrinsic—which accounts for 20% of cases—and acquired, represents an urgent medical need requiring new therapeutic approaches. The treatment of relapsed disease generally depends on the platinum-free interval (PFI), as the likelihood of response to platinum-based therapy is higher when the PFI is greater than 12 months [3]. Approved agents for relapse include gemcitabine, liposomal doxorubicin and taxane [4]. Based on the ESMO 2023 guidelines [5], platinum-sensitive patients might benefit from a combination of platinum compounds with paclitaxel, gemcitabine or pegylated liposomal doxorubicin (PLD), possibly followed by bevacizumab and PARPi; PLD and trabectedin might be an option in patients hypersensitive to platinum. In patients refractory or resistant to platinum, monotherapies paclitaxel, gemcitabine, PLD or topotecan are the best therapeutic options, although the overall response rates to these agents are low (8 to 20%) [3,6].

Polo-like kinase 1 (PLK1) is a member of the well-conserved serine/threonine polo-like kinase family with a key role in mitotic progression [7]. PLK1 is overexpressed in many types of cancer, including ovarian cancer, and its overexpression has been associated with poor patient outcome and with resistance to therapy [7,8].

Onvansertib (NMS-1286937) is a pyrazoloquinazoline and a highly specific PLK1 inhibitor in Phase II clinical development that has recently shown activity in colorectal cancer. Onvansertib has been reported to induce a mitotic cell-cycle arrest, followed by apoptosis in cancer cells, as well as xenograft tumor growth inhibition [9,10].

We demonstrated that the combination of onvansertib and paclitaxel was active in platinum-resistant ovarian cancer xenograft models [11], and the rationale for this combination was based on the fact that paclitaxel is a valid option as second-line therapy and that both drugs are active in the same phase of the cell cycle. In addition, the combination of onvansertib and olaparib has been shown to be active in olaparib-resistant preclinical models of ovarian carcinoma [12], suggesting that its combination with DNA-damaging agents may also be of therapeutic value. Recent data demonstrated onvansertib involvement in the DNA damage response pathway [13], further supporting the combination of PLK1 inhibitors with DNA-damaging agents. Given this background, we tested the activity of onvansertib in combination with gemcitabine, an analog of cytosine arabinoside (Ara-C) used as anticancer agent due to its ability to interfere with DNA synesis by suppressing DNA polymerase activity [14,15], or a second-generation platinum-based compound that covalently bind to DNA bases, forming carboplatin-DNA adducts that eventually lead to apoptosis [16], in two patient-derived xenografts of HGSOC resistant to cisplatin (DDP). We found that the combinations were well tolerated with great antitumor activity.

## 2. Results

Two PDX models were selected from our xenobank: one with acquired resistance to DDP (#266R) obtained after multiple in vivo drug treatments and the other (#315) with intrinsic DDP resistance to test the antitumor activity of onvansertib, gemcitabine, carboplatin and their combinations. As described in Materials and Methods, after randomization, mice were treated with vehicle, single agents or the combinations; onvansertib/gemcitabine and onvansertib/carboplatin combinations were well tolerated, as body weight loss never exceeded 8% and was reversible in all the cases after drug withdrawal (Appendix A). As shown in Figure 1A, all single-drug treatments were completely inactive in #266R, while the combinations of onvansertib/gemcitabine and onvansertib/carboplatin resulted in a significant increase in survival compared to controls and single drugs (*p* < 0.001 versus control, onvansertib, gemcitabine and carboplatin). In fact, the median survival time was 146 days and 154 days in the onvansertib/gemcitabine and onvansertib/carboplatin combinations, respectively, representing a 6.7- and 7-fold increase over the control median survival time (22 days).

Similar efficacy was observed in the s.c. #315 PDX model (Figure 1B); onvansertib and carboplatin monotherapies were inactive, gemcitabine monotherapy was marginally active, while both combinations were highly active. Tumor stabilization was observed throughout the 4 weeks of combined treatment and tumor weights were the lowest in the two combination groups (Figure 1B, lower panels). Tumor growth was inhibited in both combination groups for an additional 2 weeks after treatment interruption, supporting a strong cytostatic/cytotoxic effect of onvansertib/gemcitabine and onvansertib/carboplatin regimens.

To elucidate the molecular mechanisms underlying the efficacy of the combinations, we analyzed the tumors of mice treated with vehicle, single agents or the combinations for four consecutive days. Tumors were collected at 2 and 24 h after the last onvansertib treatment. We examined the induction of pSer139H2AX (γH2AX) and pSer10H3 (pH3), which are, respectively, markers of DNA damage, including apoptotic DNA fragmentation due to apoptosis, and mitotic block. In the #266R model (Figure 2A), the kinetics of γH2AX induction after gemcitabine and carboplatin monotherapy and combination treatment were different. Indeed, while after gemcitabine monotherapy a peak was reached at 2 h with resolution by 24 h, the γH2AX induction was still clearly visible in the combination-treated tumors at 24 h, suggesting the presence of more damage/apoptosis following treatment with the combination. Similarly, tumors treated with onvansertib/carboplatin showed a marked increase in γH2AX at both time points, while carboplatin monotherapy caused a mild induction of γH2AX and onvansertib monotherapy caused an induction of γH2AX at 2 h with a slight decrease at 24 h. These data suggest that more sustained damage can be observed following treatment with the combination, although not statistically significant (Appendix A). In the #315 model, the induction of γH2AX was inferred by the fact that its level was high in one of the two vehicle-treated tumors; however, its induction seemed to be higher and still visible at 24 h in both combinations as compared to single treated tumor samples (Figure 2B). Looking at pH3, a marker of mitotic block, a clear induction was observed only after onvansertib monotherapy in the #266R model (more evident at 24 h). In the #315 model, onvansertib caused a clear induction of pH3 in the single-agent-treated tumors and a partial increase in the ones treated with either of the combinations.

Taken together, these data suggest that a higher induction of γH2AX was observed with both combinations in both models. γH2AX is considered to be a marker of DNA damage, but also of apoptosis [17]. Therefore, we evaluated caspase activity in protein extracts obtained from the same tumor samples used for Western blot analyses. As shown in Figure 2C,D, the onvansertib/gemcitabine combination treatment induced a higher and more sustained level of caspase activation at 24 h compared to controls, although no statistical differences could be observed due to the intrinsic variability. In #266R model, the onvansertib/carboplatin combination treatment induced the highest level of caspase activation among the different experimental groups, while in the #315 model, caspase activation was similar in all tumor samples treated with single or combined agents.

To corroborate these data, we analyzed the positivity for both apoptotic cells and cleaved caspase-3 detection in FFPE tumor samples from untreated, single-agent- and combination-treated mice by IHC. As shown in Appendix A, in both the PDXs, an increase in the apoptotic index and the percentage of cleaved caspase-3 was observed in treated versus untreated tumors. However, due to biological variability, no clear significant difference was found between single agents and combination treatments. Interestingly, we found an increase in the percentage of necrotic cells (Appendix A) in single-agent- and combination-treated cells, especially in the #266R model; in the #315 model, this effect was less clear due to the high percentage of necrotic tumor cells in control vehicle-treated tumor samples. As a whole, these data suggest that the combination-treated tumors underwent a higher percentage of damage, eventually leading to cell death (apoptosis and necrosis in the #266R model and apoptosis in the #315).

## 3. Discussion

PLK1 has been identified as a new potential therapeutic target in cancer [7], and onvansertib is an oral, third-generation PLK1 inhibitor in clinical development in both hematological and solid tumors [10]. The manageable safety profile and promising anti-tumor activity of the combination of onvansertib/FOLIFRI/bevacizumab in the second-line treatment of patients with KRAS-mutant metastatic colorectal cancer (mCRC) was reported. The recently published Phase II trial showed that the efficacy of this combination was more marked in KRAS-mutant mCRC patients with no prior exposure to bevacizumab (exhibiting an overall response rate of 76.9 and a median progression-free survival rate of 14.9 months) [9]. In addition to these encouraging clinical data, several preclinical data suggest that the combinations of onvansertib with both cytotoxic and targeted agents are safe and of therapeutic value, with reported synergistic activities in various in vitro and in vivo cancer models [8,18,19], including our own data in ovarian cancer models [11,12].

Platinum resistance is an unmet medical need and a poor prognostic factor for ovarian cancer patients, implying that new therapeutic options need to be developed in this specific setting [20,21]. The combinations of onvansertib/carboplatin and onvansertib/gemcitabine were tested in HGSOC PDX models with both acquired and intrinsic resistance to DDP. The combinations were well tolerated and showed antitumor activities with impressive increase in mean survival time in mice bearing the orthotopic #266R model and a remarkable tumor growth inhibition in the s.c. #315 model. The inhibition of PLK1 (by shRNA and/or small inhibitors) has been reported to enhance the cytotoxic activity of cisplatin and antimetabolic agents [21,22], but few data have been reported in the setting of platinum resistance [23,24]. All three drugs have dose-limiting toxicological effects [25]; however, the striking antitumor activities observed and the lack of observable [13] toxicities in mice suggest that these combinations can be safely translated into a clinical setting. The great anticancer effect suggests the possibility to lower the dose while maintaining the efficacy. The molecular mechanism underlying the efficacy of the combinations suggests a higher induction of DNA damage, which seems plausible considering that, in both cases, gemcitabine and carboplatin, respectively, interfere with DNA metabolism [15] and induce alkylation damage [16]. Indeed, our data suggest a more sustained induction of damage and cell death (apoptosis and/or necrosis) following gemcitabine/onvansertib and carboplatin/onvansertib combinations. Even if the precise mechanisms at the basis of the observed additive/synergistic activities have not been elucidated, the inhibition of PLK1 functions in different DNA repair pathways involved in the gemcitabine- and carboplatin-induced DNA damage and/or the role of PLK1 in the G2/M transition can be hypothesized [7]. The inhibition of repair could increase in DNA damage to a level leading to cell death; alternatively, carboplatin and gemcitabine have been reported to cause a block in the G2/M phases of the cell cycle which could add to the one induced by PLK1 inhibition, thus inducing a mitotic catastrophe. Considering that the induction of pH3 was only clearly visible in the onvansertib-treated sample, we would favor the first hypothesis.

## 4. Materials and Methods

Animals and drugs. Five-week-old female NCr-nu/nu mice were obtained from Envigo Laboratories (Bresso, Italy) and maintained under specific pathogen-free conditions. Procedures involving animals were conducted in conformity with the following laws, regulations, and policies governing the care and use of laboratory animals: Italian Governing Law (D. lg 26/2014; authorization no.19/2008-A issued 6 March 2008 by the Ministry of Health); Mario Negri Institutional Regulations and Policies (Quality Management System Certificate: UNI EN ISO 9001:2015, reg. no. 6121); the NIH Guide for the Care and Use of Laboratory Animals (2011 edition); and EU directive and guidelines (EEC Council Directive 2010/63/UE). The in vivo experiments on PDXs were approved by the Italian Ministry of Health (approval no. 646/2022-PR). Tumor fragments (MNHOC315, #315) were subcutaneously (s.c.) transplanted, and tumor cell suspensions (10^7^cells/mouse for MNHOC266R, #266R) were orthotopically implanted (intraperitoneally, i.p.). #315- and #266R-bearing mice were, respectively, randomized when tumors reached approximately 100–150 mg or 10 days after i.p. transplant to different experimental groups (control; onvansertib—40 mg/kg 5 days a week for 4 weeks-; gemcitabine—60 mg/kg once a week for 4 weeks-; carboplatin—50 mg/kg once a week for 4 weeks-; and their combinations). Tumor growth was measured twice weekly with a Vernier caliper, and tumor weights (mg = mm3) were calculated as follows: (length (mm) × width^2^ (mm^2^))/2, where width < length. For i.p. transplanted tumors, mice were weighted three times a week. For pharmacodynamics studies, tumor-bearing mice were treated for 4 days at the same dose and frequency as for the efficacy studies, and tumors were collected 2 and 24 h after the last dose of onvansertib for downstream analyses.

Onvansertib was dissolved in DMSO 10% and vehicle solution (0.5% methylcellulose and 1% Tween-20) and prepared freshly; carboplatin and gemcitabine (Merck Life Science, Darmstadt, Germany) were dissolved in NaCl 0.9%.

Western blot analysis. Snap-frozen tumor fragments were lysed in ice-cold cell extract buffer containing 50 mM TrisHCl pH 7.4, 250 mM NaCl, 0.1% Nonidet NP40, 5 mM EDTA and NaF 50 mM with a protease inhibitor cocktail (Sigma-Aldrich, St. Louis, MO, USA). Protein (40  µg) was resolved on 12% SDS-PAGE gels and transferred to nitrocellulose membranes (Merck Millipore, Burlington, MA, USA), and immunoblotting was carried out with the following antibodies and visualized using Odyssey FC Imaging System (Li-COR, Lincoln, NE, USA): anti-βactin (sc-47778, Santa Cruz Biotechnology, Dallas, TX, USA); anti-phospho-Histone H3 (Ser10) (6G3) (#9706, Cell Signaling Technology, Danver, MA, USA); anti-phospho-H2AX (pSer139) (#9718, Cell Signaling); anti-rabbit and anti-mouse (#1706515, #1706516, Bio-Rad Laboratories S.r.l. Hercules, CA, USA) secondary antibodies.

Caspase-3 activity assay. Caspase-3 activity was measured by an enzymatic assay using the Caspase-Glo^®^3/7 kit (Promega Corporation, Madison, WI, USA) following the manufacturer’s instructions. Tumor protein extracts were dispensed on a white 96-well plate and incubated at room temperature for 45 min; then, luminescence was read using a plate reader (GloMax Discover, Promega Corporation). Caspase activity was expressed as mean relative light units (RLU) normalized to the protein concentration [12].

All the immunohistochemical (IHC) methods are specified in the Appendix A.

Statistical analysis. Statistical significance was determined with GraphPad Prism 7.05 (GraphPad Software). Figure legends specify which test was used.

## 5. Conclusions

The present data suggest that the combinations of onvansertib with gemcitabine and carboplatin are very effective in ovarian carcinoma with both intrinsic or acquired platinum-resistance, probably due to higher DNA damage accumulation that leads to cell death both by apoptosis and necrosis. Of note, both combinations were proved to be safe in our in vivo models, fostering their clinical translatability.

## Figures and Tables

**Figure 1 ijms-26-05708-f001:**
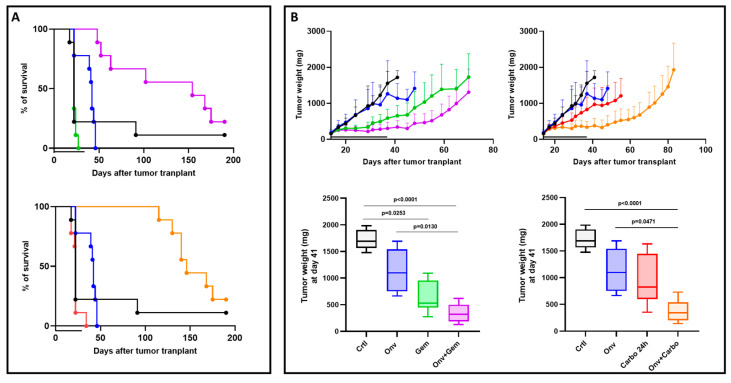
Antitumor activity of single agents and combinations in #266R and #315 HGSOC PDXs. Panel (**A**). Kaplan–Meier survival curves of mice transplanted with #266R. Mice transplanted intraperitoneally with #266R xenograft were randomized to receive vehicle (black), onvansertib (blue), gemcitabine (green), carboplatin (red), onvansertib + gemcitabine (violet) or onvansertib + carboplatin (orange). Log-rank Mantel–Cox test. Panel (**B**). Upper panels. Tumor growth: #315 xenografts were transplanted subcutaneously and when tumor masses reached 100–150 mg, mice were randomized to receive vehicle (black), onvansertib (blue), gemcitabine (green), carboplatin (red), onvansertib + gemcitabine (violet) or onvansertib + carboplatin (orange). Lower panels. Tumor volumes at the end of the treatment period (4 weeks). Data are the mean ± SD of tumor masses and each group consisted of 8–10 animals.

**Figure 2 ijms-26-05708-f002:**
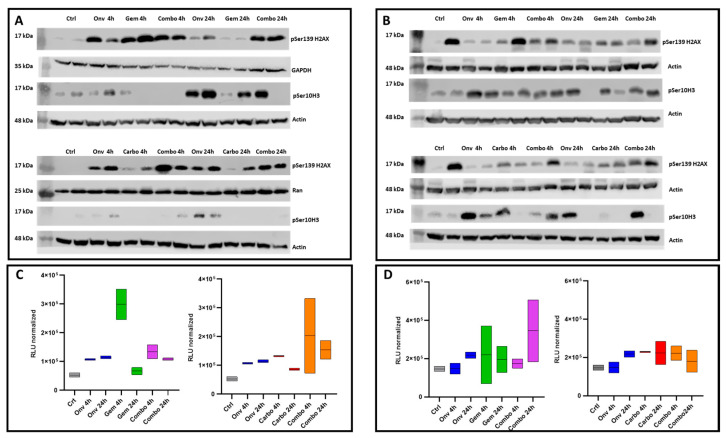
In vivo pharmacodynamic assessment of mitotic block, DNA damage and apoptosis in PDXs treated with vehicle, single agents or combinations. Western blot analysis showing pSer10 H3 (pH3) and pSer139 H2AX (γH2AX) protein levels in tumor protein extracts from #266R (panel (**A**)) and #315 (panel (**B**)) xenografts. Caspase-3 activity in tumor protein extracts from #266R (panel (**C**)) and #315 (panel (**D**)) xenografts.

## Data Availability

Data are available upon request to the corresponding author.

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
