# Peer review of "Onvansertib-Based Second-Line Therapies in Combination with Gemcitabine and Carboplatin in Patient-Derived Platinum-Resistant Ovarian Carcinomas"

_ijms, 2025, doi:10.3390/ijms26125708_

Round 1
Reviewer 1 Report
Comments and Suggestions for Authors
Guffanti et al. investigated the therapeutic efficacy of Onvasertib as a monotherapy and in combination with standard chemotherapeutic agents (gemcitabine or carboplatin) and using two PDX models of platinum-resistant ovarian cancer. The single agent or the combinations with Onvasertib and SOC were well tolerated in the PDX mice. The combination therapy resulted in favorable probability of survival and reduced tumor growth. The authors then analyzed the molecular pathway of drug efficacy by measuring the induction of DNA damage and mitotic bloc (gH2AX and pH3). The Ovasertib combination with both drugs induced DNA damage in both PDX models suggesting the therapeutic value of combining Ovasertib in the management of platinum-resistant ovarian cancers.
The research presented is relevant for the transferability of results since it uses PDX models where two types of resistance are included, however the limitation of using only 2 models is the context of heterogeneity characteristic of OVC. The study is also limited by the experimental set-up. Data should be validated by other means such as IHC of collected tumors.
Reviewer 2 Report
Comments and Suggestions for Authors
The chemical structures and detailed modes of action of carboplatin, Onvansertib, and gemcitabine should be added to the introduction part.
The introduction part should include detailed mode of action and role of PLK1 in carcinogenesis and in which cancers it was overexpressed.
The manuscript is missing conclusion part.
Round 2
Reviewer 1 Report
Comments and Suggestions for Authors
The authors have provided a minimal analysis of IHC data, partially confirming their hypothesis. Is there DNA damage? The discussion of the data should be implemented, as well as representative images should be provided.
Author Response
Comments 1: The authors have provided a minimal analysis of IHC data, partially confirming their hypothesis. Is there DNA damage? The discussion of the data should be implemented, as well as representative images should be provided.
Response 1: The IHC data added were apoptosis, necrosis and cleaved caspase-3. Specifically histological sections stained with H&E were examined with a light microscope for the detection of lesions and for the quantification of: 1) the necrosis (expressed as percentage of the necrotic area(s) in relation to the total area of the section) and 2) the apoptotic index (based on the number of apoptotic figures detected in three randomly selected high power microscopic fields). Blinded evaluation of histological slides was carried out (i.e., without information about the experimental groups). Digital Image Analysis was performed quantify the number of Cleaved Caspase-3 positive as specify in Supplementary Materials. We focused on these pharmacodynamic read outs as we reasoned that the western data on the higher and sustained induction of gH2AX in tumor extracts were quite convincing. The data demonstrated that in the case of #266R model higher cell death (necrosis and apoptosis) could be observed in the combo group, even if it did not reach the statistical significance likely due to tumor heterogeneity. However, if the referee and the editor think that H2AX by IHC is needed we will do.
As suggested by the referee, we have better discuss these data in the results and discussion sections: In addition, we have also included in Figure S3, the IHC of cleaved caspase-3. If the referee thinks we have also to include the IHC of necrosis we will.
Reviewer 2 Report
Comments and Suggestions for Authors
The authors responded adequately to the comments and the manuscript is now suitable for publication in the current revised form
Author Response
Comments 1: The authors responded adequately to the comments and the manuscript is now suitable for publication in the current revised form
Response 1: We thank the referee.
Round 3
Reviewer 1 Report
Comments and Suggestions for Authors
Authors have added low resolution images of IHC staining, with such resolution is not possible to draw any conclusion. Surprisingly the quantifications presented in the previous version have been removed.
Author Response
Comments1: Authors have added low resolution images of IHC staining, with such resolution is not possible to draw any conclusion. Surprisingly the quantifications presented in the previous version have been removed.
Response1: We have now provided higher resolution images of IHC staining.We did not remove the quantification data, but changed the graph axes names, we changed them again and we hope that now the data are clearer. For clarity we did two figures, one with the quantification of the apoptosis and necrosis and the corresponding IHC images in tumor samples of PDX #266R (Supplementary Figure S3) and one in tumor samples of PDX #315 (Supplementary Figure S4).